# Holistic chemical evaluation reveals pitfalls in reaction prediction models

**Victor Sabanza Gil**[123*]   **Andres M. Bran**[12*]   **Malte Franke**[1*]
**Rémi Schlama**[1]   **Jeremy S. Luterbacher**[23]   **Philippe Schwaller**[12]
[1] Laboratory of Artificial Chemical Intelligence (LIAC), ISIC, EPFL
[2] National Centre of Competence in Research (NCCR) Catalysis, EPFL
[3] Laboratory of Sustainable and Catalytic Processing (LPDC), ISIC, EPFL
[*] Contributed equally.
philippe.schwaller@epfl.ch

## Abstract

The prediction of chemical reactions has gained significant interest within the machine learning community in recent years, owing to its complexity and crucial applications in chemistry. However, model evaluation for this task has been mostly limited to simple metrics like top-k accuracy, which obfuscates fine details of a model's limitations. Inspired by progress in other fields, we propose a new assessment scheme that builds on top of current approaches, steering towards a more holistic evaluation. We introduce the following key components for this goal: CHORISO, a curated dataset along with multiple tailored splits to recreate chemically relevant scenarios, and a collection of metrics that provide a holistic view of a model's advantages and limitations. Application of this method to state-of-the-art models reveals important differences on sensitive fronts, especially stereoselectivity and chemical out-of-distribution generalization. Our work paves the way towards robust prediction models that can ultimately accelerate chemical discovery.

## 1  Introduction

In recent years, there has been a significant increase in the development and application of machine learning (ML) algorithms for solving various tasks in science-related fields[1–5]. Advances in these models have been greatly accelerated by model developments[6–8], acquisition of extensive training data[9,10], and the establishment of benchmarks[11–17], which have enabled researchers to evaluate and compare new models based on multiple aspects relevant to the task at hand. Chemistry has also experienced remarkable progress in problems such as retrosynthetic planning[2,18–23], reaction condition recommendation[24], reaction prediction[25–29], and others[30–36]. Among these, reaction prediction has gained considerable importance due to its broad applicability in areas such as waste material valorization[37], reaction network analysis[38], and even the evaluation of retrosynthesis prediction models[39]. Compared to the other tasks, reaction prediction benefits from having a less ambiguous objective, simplifying the evaluation process.

A wave of progress in this field has been further propelled by the publication of the USPTO reaction dataset[9,40,41], which has led to the emergence of benchmarks for various tasks, including USPTO_STEREO[42] and USPTO_480k[43] for reaction prediction. These consist of tailored subsets of the USPTO dataset, randomly split for training and evaluation. From the algorithmic side, transformer-based sequence-to-sequence models have emerged as the top-performing algorithms for reaction prediction[44], achieving top-1 accuracies of over $91\%$[29] on stereochemistry-free datasets. Other widely used model types include template-based[45] and graph-to-sequence models[46], each

leveraging different inductive biases derived from chemical expertise, achieving comparable top-1 accuracy.

Evaluation has received considerable attention in fields such as computer vision[47,48] and language models (LMs)[49,50]. Out-of-distribution (OOD) shifts have been thoroughly discussed[51,52], and the need for testing in this domain has been emphasized[53]. Broader evaluation schemes that aim to expose the strengths and failure modes of different models have also been proposed for LMs[54]. However, standardized model evaluation in the field of reaction prediction has been largely neglected, with most studies relying solely on top-k reaction outcome accuracies, a restricted measure of model performance that overlooks a diverse range of complexities inherent to reaction prediction. Some works perform additional analyses and comparisons, giving more insight into the model's performance, however they lack a standardized format and are constrained by the quality of the reaction data that is used[55,56].

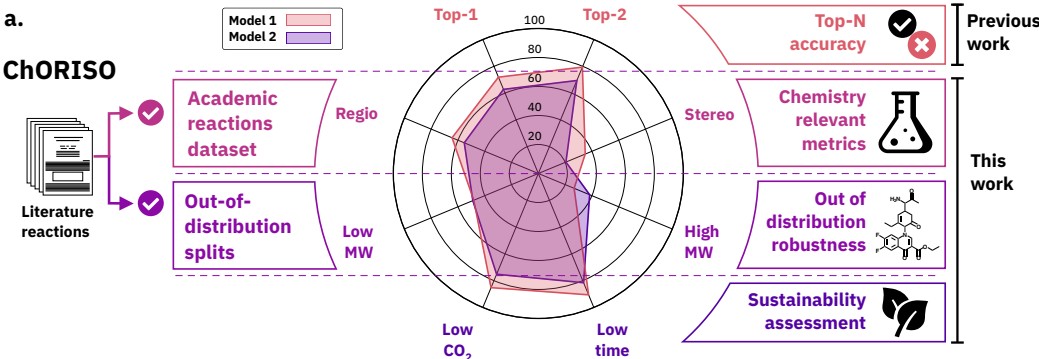

Figure 1: **Holistic evaluation of reaction prediction ML models. a.** Chemistry-relevant metrics, out-of-distribution (OOD) robustness tests and sustainability assessments are proposed. Evaluation is done by using the CHORISO dataset, which provides reaction entries and from which OOD splits are derived. **b.** Failure modes of current reaction prediction models include poor performance in reactions with potential selectivity issues (left), and OOD scenarios such as reactions involving species with higher molecular weight (right).

In this work, we propose a holistic evaluation pipeline for a better chemical assessment of reaction prediction models. For this purpose, we introduce CHORISO (**CH**emical **O**rganic **R**eact**I**on **S**MILES **O**mnibus), a curated dataset of academic chemical reactions, along with a suite of chemically relevant metrics for the standardized evaluation of these models. Addressing the limitations in existing evaluation methods, we propose multiple slices and splits of CHORISO, serving as distinct scenarios fortesting, considering OOD scenarios[51]. Several chemically relevant desiderata have been implemented in the proposed standard metrics, including different types of chemical selectivity, along with measures of model efficiency and environmental impact, as shown in Figure 1a. Through a combination of standardized metrics, curated data, OOD testing, and a collection of models, we aim to facilitate the development new cutting-edge ML models for reaction prediction (Figure 1b). This effort could ultimately lead to more accurate and reliable predictions with applications in various fields.

## 2 Holistic Evaluation of Chemical Reaction Models

Prediction of chemical reaction outcomes is a key problem in organic chemistry. Achieving accurate, trustworthy, and scalable chemical reaction prediction could accelerate chemistry discovery[37,57].

Excellent performance in this task is thus crucial for the development of chemical models in general. These models are typically tested and compared in the literature using the top-k accuracy —with k $\in [1,5]$, with the best models reaching top-1 accuracies higher than 91% on patent reaction benchmarks without stereochemical information[46]. However, the utility of these models is limited, as they tend to underperform in real-world use cases[28,55]. The disconnection between such high accuracies and poor practical performance thus highlights the need for better evaluation methods.

The recent work of Liang et al.[54] sets the basis for the Holistic Evaluation of Language Models (HELM). The authors argue that, given the flexibility and generality of LMs, it is desirable to establish an evaluation scheme that more transparently assesses the capacities and flaws of these types of models. Holistic evaluation requires the identification of potential scenarios —encoding use-cases— and metrics —encoding desiderata— that are of interest to LMs. In this setting, the authors can adequately test the models in terms of accuracy, robustness, and toxicity, among other metrics, across a wide range of scenarios, exposing the advantages and trade-offs of popular LMs. Building upon this work, our aim in this section is to identify relevant settings where chemical reaction models need to be tested and to develop metrics that align with key desired characteristics of these models.

## 2.1 Data

A critical piece on the road toward holistic evaluation is data. Data not only feeds the models with latent knowledge but also allows researchers to model scenarios for testing, ultimately allowing to gauge and compare the adequacy of reaction prediction models in such scenarios. Despite the wave of reaction prediction models fueled by the USPTO dataset[41,58], these models learn from regions of the chemical space that are not necessarily typical targets of academic researchers, mostly featuring well-established, industry-relevant reactions.

To alleviate this, we propose CHORISO, a curated reaction dataset of diverse academic reactions. CHORISO is a mix between cleaned and processed versions of CJHIF, a dataset of reactions extracted from high-impact academic journals[59], and USPTO, a dataset of reactions extracted from patents[9] (see Appendix A.1). As shown in Figure 2, CHORISO features around 2.2M reactions, including a high ratio of C-C bond formation and functional group interconversion reactions, which are fundamental for strategic synthetic planning[60]. CHORISO additionally exhibits heavier-tailed distributions of molecular weight and number of stereocenters in products, covering a larger and more relevant portion of the chemical space. This is particularly important for applications where stereocontrol is fundamental[61], as well as for investigations regarding the scope of chemical reactions, where extrapolation to higher Product's Molecular Weight (PMW) —e.g. larger substituents— is desired. We use this dataset as the data source for our holistic evaluation.

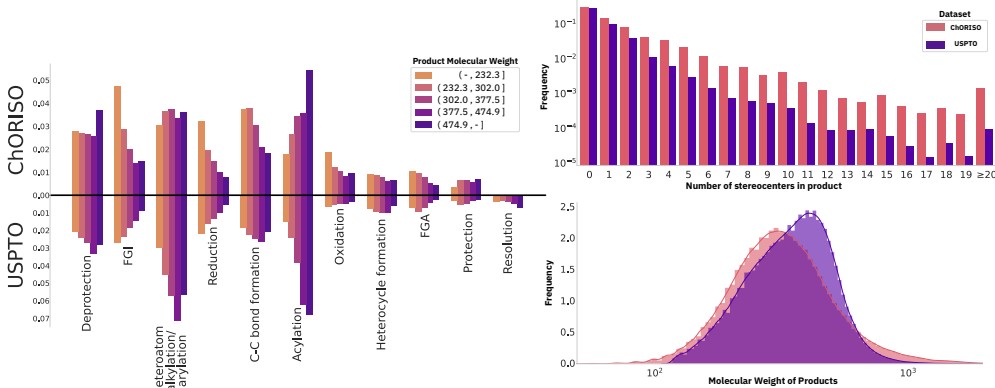

Figure 2: **Dataset distributions comparison.** Differences between the two datasets (CHORISO and USPTO) are highlighted. **a.** Reaction types across various bins of product molecular weight (PMW). The distribution of reaction types varies considerably across PMW for CHORISO. Unrecognized reactions in the datasets are not plotted for clarity. **b.** Distribution of number of stereocenters in products shows that products in CHORISO overall contain a higher number of chiral centers. **c.** PMW distribution. CHORISO's distribution makes it suitable for OOD splits on both ends of the PMW distribution.

## 2.2 Desiderata for models

A number of properties are expected from reaction prediction models. High accuracy is certainly one of them, but over-reliance on this metric can be misleading, especially in unbalanced datasets[55]. From the chemical side, good handling of stereo- and regiochemistry is desired, as well as a correct understanding of functional group effects, effects of catalysts, reagents and other additives, and conditions like temperature and pressure[55]. Proper modelling of these variables is necessary to have an accurate prediction system, and measuring how models perform in each variable is central to identifying trade-offs between them. Other, less field-specific properties are also desired, such as low inference speeds, low carbon footprint, and low energy consumption, which become important in high-demand applications like chemical networks space exploration[57].

In this work, we pave the way towards tackling the accurate evaluation of models in these regards, by implementing two chemically relevant and two performance-related metrics. The first two, regio- and stereo-accuracy, calculate the product prediction accuracy in subsets of reactions that are flagged to have regio- and stereo-selectivity issues. These reactions are commonly encountered in aromatic ring substitutions[62] or systems where chirality is affected during the reaction[28], among others[63]. These metrics thus highlight the models' capacities to predict the most reactive sites in a given context correctly, and to predict the dominant isomer in a potential isomer mixture. Furthermore, measurement of $CO_2$ emissions and training time are also considered, as per raising sustainability concerns in AI[64]. Addressing sustainability in models can also improve their accessibility due to reduced hardware requirements for model execution during inference.

## 2.3 Scenarios

Identifying and recreating scenarios relevant for testing models is one of the two key points highlighted by Liang et al.[54]. We pay particular attention to out-of-distribution (OOD) scenarios, where the train and test set distributions differ. Due to the inherent difficulties in characterizing feature relevance for this task, we focus mainly on marginal shifts, described by Teney et al.[51] as those where a distribution shift happens only across features irrelevant to the task. One easily measurable property, that to good approximation is irrelevant for reaction prediction, is the product molecular weight (PMW). A model will desirably perform well independent of the molecular size, as reactivity analysis is typically based on local molecular features such as functional groups, which PMW does not directly influence.

Following this reasoning, two types of data splits are proposed: low PMW and high PMW, where the test data corresponds to the lowest and highest end of the PMW distribution, respectively (Appendix A.4). In addition, a standard split by products is proposed, where the set of product molecules in the train set is disjoint from its counterpart in the test set. These sets are used to evaluate the model and to provide a wider picture of its capabilities. While we acknowledge the importance of other types of distribution shifts, and future research should focus on exploring these, our current approach already proves valuable in revealing several aspects of models, including their limitations and failure modes, as shown in Section 3. This demonstrates the importance and effectiveness of the scenarios and distribution shifts considered in our study.

## 3 Results and discussion

With the proposed holistic evaluation pipeline, two high-performing reaction prediction models from the literature were trained and evaluated. The models, Graph2SMILES[46] (G2S), and Molecular Transformer[27] (MT) both draw elements from the Transformer architecture[7]. This architecture relies on the attention mechanism[65] to infer inter-dependencies between tokens in token sequences. While the MT uses a string representation as input and output[44], G2S uses a graph encoder with a string decoder. Figure 1a provides a general comparison of the models, evaluated as proposed in this work using the CHORISO dataset. Contrary to previous reports[46], top-k accuracy indicates an advantage of MT over G2S. However, extending from this, our approach reveals important differences in models, namely their distinct performances in stereoselectivity and OOD generalization, both key for an appropriate evaluation at the chemical level. Figure 1 also illustrates how the implementation of the proposed holistic evaluation pipeline allows to dissect a model into multiple performance factors, providing a better picture of model limitations and trade-offs.

The models in question were trained and tested on multiple splits of the CHORISO dataset. These splits consisted of a random split, a split by products, and two splits by PMW (see Appendix A.4) to measure out-of-distribution performance. As displayed in Table 1, in-distribution splits show advantage of MT over G2S. Notably, results on the random split are systematically higher than those on the product split, highlighting the more challenging nature of the latter, and supporting previous claims of random splits leading to misleading results for generalization[55]. The product split is thus kept as the standard split for this work and all the top-k and selectivity comparisons are based on the predictions on this set. At a high level, top-k accuracy shows lower values compared to the reference reported values. The same training procedure and evaluation on the full USPTO dataset reveal a similar trend on this dataset (Appendix B). The source of the general performance decrease could be assigned to the inherent more difficult split by product and to the nature of both datasets (bigger and noisier than the original USPTO_480k that was used in the original work). Prior studies demonstrated that Transformer models typically achieve 60-70% top-1 accuracy on noisy datasets, consistent with the results for CHORISO and USPTO in this study[44,66]. Finally, a high top-1 accuracy difference (around 10%) between MT and G2S on the CHORISO dataset compared to USPTO also suggests that MT adapts better to the different chemical reactions contained in the new CHORISO dataset.

| Split type | Model | Top-1 acc (%) | Top-2 acc (%) | Stereo-acc (%) | Regio-acc (%) |
|---|---|---|---|---|---|
| Product | MT | **71.9** | **79.4** | **35.0** | **64.4** |
|  | G2S | 62.8 | 69.4 | 20.6 | 55.3 |
| Random | MT | **72.6** | **80.2** | **37.7** | **64.9** |
|  | G2S | 62.9 | 69.6 | 22.2 | 54.5 |
| low PMW | MT | 48.2 | **57.0** | **23.5** | **34.8** |
|  | G2S | **48.9** | 56.6 | 20.3 | 31.7 |
| high PMW | MT | 26.7 | 29.8 | 12.8 | 24.7 |
|  | G2S | **33.6** | **37.4** | **17.0** | **32.8** |

Table 1: **Model benchmarking results.** Performance of Molecular Transformer (MT) and Graph2SMILES (G2S) models for each split of the CHORISO dataset. The columns display the results of each model/split combination for each of the proposed metrics. Bold font highlights the best performing model in a metric, for a given data splitting method.

Chemistry-specific metrics enrich the analysis and reveal important references between both models. Notably, the performance of both models in selectivity metrics decreases relative to top-k accuracy, an expected behavior given that these scores are computed by selecting an inherently more challenging subset of the test data. Apart from representing a smaller subset of the main test data, regioselective and stereoselective reactions impose an extra layer of difficulty on the reaction prediction task (see Appendix A.3. In these cases, the model has to learn the preferred site reactivity or product chirality in addition to the main task of learning the reactive pattern based on molecule functional groups. MT outperforms G2S both in terms of regio and stereo-accuracy. In terms of regiochemistry, the performance difference between MT and G2S is similar the one for top-1 accuracy, suggesting that the source of advantage of MT for this type of reactions is similar than the one for the non-selective ones. However, stereo-accuracy reveals an extra performance of the MT over G2S, suggesting a special advantage of MT with respect to G2S when predicting stereoselective transformations. This fact has been noted and exploited in previous reports[28]. Likely due to the graph encoder used in G2S, this model's stereochemistry performance is substantially lower, with an over 15% loss in stereo-accuracy relative to MT in the product split. MT is thus a better performing model for reactions where product chirality plays a key role, such as those with stereocenter inversions and stereocenter formation as displayed in Figure 3. These results highlight the potential of the proposed evaluation scheme to reveal otherwise hidden differences between models, otherwise diluted in the top-k accuracy.

Thanks to our proposed metrics, we can locate and rationalize the prediction failure of a model using a chemical context. This and sources of challenge for , and not simply consider it as a merely wrong prediction. Focusing on reactions where MT performs better than G2S, we can observe some of the trends and explain the difference in performance between the models. Figure 3 displays some

reactions where MT provides the correct product and G2S fails from the selective reactions set. Reaction a) shows a stereoselective reduction using sodium borohydride where both models predict the correct molecule (in which the ketone has been reduced to an alcohol), but G2S misses the correct chirality of the generated stereocenter. Reaction b showcases a Curtius rearrangement that preserves the original chirality of the reacting center after the transformation. Here G2S also predicts the correct reacting pattern (even the stereochemistry preservation of the reaction center), but predicts a different isomer where two non-reacting chiral atoms have been inverted. Finally, in terms of regiochemistry, reaction c shows how MT predicts the correct regioisomer of a Friedel-Crafts acylation, whereas G2S generates the incorrect isomer where the acylation happens on the less activated aromatic carbon on the reacting ring. These examples showcase limitations and differences between the two methods, and may help to propose model architecture improvements to correct the selectivity difference.

Figure 3: **Model limitations revealed by chemistry-specific metrics.** MT performance is better than G2S in reactions where stereocenters are formed or different regioisomers are possible. **a.** Example where MT predicts the correct product and G2S fails because it predicts the opposite chirality in the reacting center **b.** Example where MT predicts the correct product and G2S fails because it inverts the chirality of non-reacting atoms. **c.** Example where MT predicts the correct product and G2S generates a different regioisomer.

An opposite trend is observed in the OOD splits (low and high PMW), where G2S outperforms MT with differences of up to 7% top-1 accuracy in high PMW. As shown in Figure 4, this difference is further exacerbated as the PMW increases, indicating the strong sensibility of MT to distribution shifts. When analyzing the accuracy by MW split, MT has an initial performance advantage over G2S (higher top-1 accuracy for the reactions where PMW is below or above 100 g/mol to the PMW of the training reactions). MT performance drastically decreases when moving away from training PMW, especially in the high PWM split. G2S is, on the other hand, more robust, as shown by the lower rate of decay in Figure 4. This behavior is hypothesized to be due to the graph-based encoder of G2S, which potentially suffers less from shifts in the molecular size of the input reactants by focusing on local molecular features. The MT instead suffers more from such distribution shifts as PMW directly affects input sequence length. This issue becomes more important for longer sequences, a fact that has been documented previously for language models[67,68] and that can be reflected on the poorer performance of MT on the high PWM. In the out-of-distibution scenarios, MT thus tends to predict larger molecules in the lowPMW split, and smaller molecules in the highPMW split, as shown in Figure 4. In addition, the general performance of both models decreases in the ODD scenario, suggesting that this distribution shift can be used to propose architecture improvements that make models less susceptible to this shift in non-relevant features. Finally, these results highlight the strengths of graph-based models and G2S in particular, which make them a more robust option for scenarios where a shift in property distribution like PMW is expected.

A final perspective based on model efficiency is provided by the sustainability metrics. Detailed analyses of each model's $CO_2$ production and training and inference time are possible thanks to

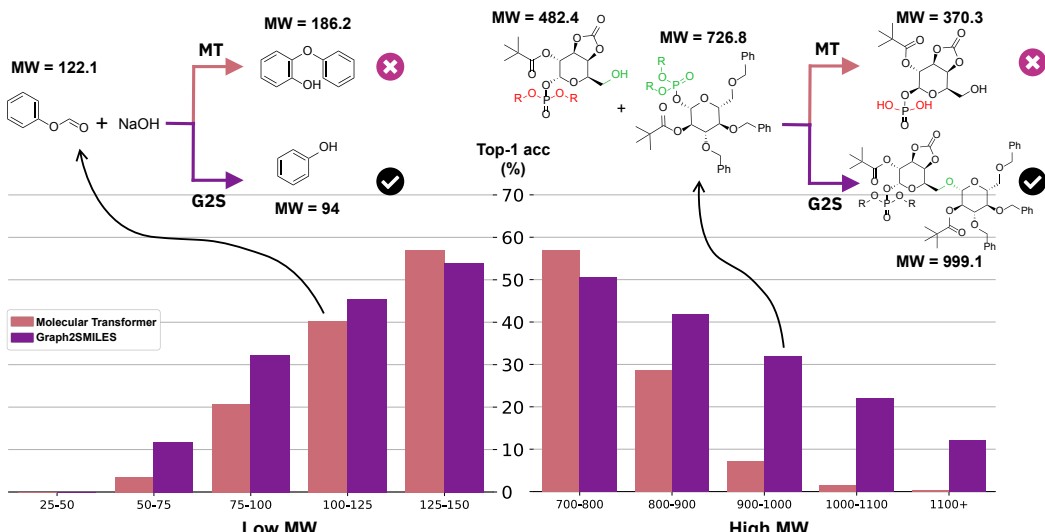

Figure 4: **Model limitations revealed by OOD metrics.** Analysis of model predictions in OOD setting shows that MT's accuracy decreases abruptly as PMW increases, while G2S is more nuanced, showing advantages in OOD generalization. Selected reactions show how MT tends to predict products with PMWs that are within the range of the training data, while G2S is generally better at extrapolation. In green, the reaction center is highlighted, showing how MT predicts the correct reaction, whereas G2S selects an incorrect transformation that lowers the MW of the resulting product and gives an incorrect prediction.

recent tools[69] developed as per recent sustainability concerns of computational research, especially in ML and AI[70]. MT produced 0.32 kg of $CO_2$ and took 19.8 h to train on the CHORISO data, whereas G2S produced 0.57 kg $CO_2$ and 40.8 h to train (full training and inference consumption for the benchmarking in Appendix B.2). The higher consumption and training time of G2S renders it less environmentally friendly than the MT, however, the environmental impact is still far less than models in other fields[71]. On the other hand, inference time is similar for both models (3.1 vs 3.7 h). Overall, MT is a more efficient model for this task compared to G2S. This metric may help orient model selection considering a possible sustainability budget for model training and evaluation.

It must be stressed that the goal of holistic evaluation is not to determine the absolute best method among all the available models, as evaluation with top-k accuracy would. Instead, the objective is to provide a detailed map of the strengths and weaknesses of each model, ultimately producing a guide into each model's scope and applicability. Therefore, we have not performed hyperparameter tuning of the models, and instead used previously reported parameters to compare general accuracy. As it has been mentioned, both models were trained for the same number of steps for a fair comparison. As increasing model accuracy was not the main goal of the work, the performance values may be lower than others found in previous reports. However, our results gave a complete comparison in terms of chemistry applicability, robustness, and sustainability of MT and GS2. Following this, MT is a better choice for stereochemically challenging reactions or a limited computational budget, whereas G2S is recommended for scenarios where the novel reaction products fall far from the training property distribution. Furthermore, these efforts will help orient researchers toward addressing specific aspects of models, leading to increasingly better models across multiple chemically relevant directions.

## 4 Conclusion

We have introduced a new holistic evaluation method for chemical reaction prediction models. This work aims to improve current model evaluation practices in chemistry, allowing a stronger assessment of their real capabilities. Following advances in other fields, we discuss and implement a set of evaluation metrics and scenarios relevant to the task of reaction outcome prediction. In addition, a new academic reactions dataset is released —CHORISO, that is better suited for recreating some of these scenarios, as compared to other existing benchmarks. Leveraging this approach allowed

us to compare two state-of-the-art reaction prediction models, revealing pitfalls and trade-offs in the models, as well as limitations in previous evaluation methods. In particular, holistic evaluation suggests that the Molecular Transformer is better suited for stereochemically challenging reactions, while requiring a fraction of the energetic budget. On the other hand, Graph2SMILES showed much stronger performance in certain out-of-distribution scenarios. These results can easily be rationalized in terms of the models' architecture, with graph-based methods generalizing better to larger graphs, while text-based methods encode spatial features like stereochemistry better. More importantly, the results out-of-the-box reveal key features from models, that are hindered by the commonly used top-k accuracy.

Overall, this holistic evaluation proposes an improved pipeline for thorough reaction prediction model evaluation. Further development is required in the design of richer chemistry-relevant metrics and the identification of additional marginal out-of-distribution splits. In spite of this, this methodology already shows great potential for evaluating models, and opens the way towards the definition of functional guidelines for enhanced model development and selection in chemistry. Selection and improvement of the best-performing models for specific types of reactions and chemical spaces would unlock their routinary application and leverage the current low-data regime. This would lastly enable a future AI-accelerated chemical research.

## Data & Code availability

All the data and code used in this work is made freely available. The CHORISO dataset, along with the train and test splits described in this paper can be found at `https://figshare.com/s/5e57a3399c52701cbc15` (DOI: 10.6084/m9.figshare.22598230). The code used for data pre-processing and analysis, metrics, and model evaluation, can be found at `https://github.com/schwallergroup/CHORISO` (data processing, analysis and metrics) and at `https://github.com/schwallergroup/CHORISO-models` (benchmarking).

## Acknowledgements

This publication was created as part of NCCR Catalysis (grant number 180544), a National Centre of Competence in Research funded by the Swiss National Science Foundation. V.S.G acknowledges support from the European Union's Horizon 2020 research and innovation program under the Marie Skłodowska-Curie grant agreement N° 945363.

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

# A  Data

Recently, Jiang et al.[59] published a dataset containing 3.2M organic reactions mined from high-impact journals (CJHIF). Each entry contains a reaction SMILES together with the reported yield, as well as the reagents, solvents, and catalysts used, all given in natural text (i.e. no machine readable formats).

## A.1  Data curation

For the cleaning and curation of the CJHIF dataset, we followed the pipeline described in Figure 5. In a first step, all the names of reagents, solvents and catalysts are extracted as natural text from the CJHIF dataset and counted by occurrence. This allowed us to translate individually each name to its corresponding SMILES string. To do that, we used Py2OPSIN[40], and PubChem API[72] in case the name could not be translated by the former. Once this translation dictionary is obtained, we analysed more deeply the names that could not be translated to SMILES, and corrected them manually based on the highest occurrence to the lowest. In addition to the manual correction by occurrence and in order to get high quality data on stereochemical reactions, we also looked for specific symbols in the names such as "+", "-", "r", "s", and manually corrected them by occurrence. These characters are normally present in chiral catalysts and are often present in stereoselective transformations. Since we still noticed important compounds that were not translated because of typos in the names or additional characters, we clustered the remaining names based on string similarity using the package fuzzywuzzy in combination with a DBSCAN algorithm provided in the scikit-learn library[73]. Once a quasi-complete translation dictionary is obtained, the compounds names in the CJHIF dataset were translated to SMILES, to form a set of full reaction SMILES. The resulting reactions are filtered to keep only those where no species were lost during translation. This guarantees a one to one correspondance and ensures the fidelity of the resulting reaction SMILES with respect to the original entry. A second filter is also applied to ensure that the products contain at most fewer atoms than the reactants, and that no atom appear in the product that was not in the reactants (to avoid uncomplete entries). A third filter is used to check if the products have any stereocenter that the reactants do not have. These reactions are then modified to remove any stereocenter, in order to keep the chemistry associated. Naturally, a catalyst with axial chirality cannot be encoded as SMILES, so we believe that to be consistent for the transformer model, stereocenters in SMILES have to be induced from the reactants SMILES. Reactions are then canonicalized, and duplicates are removed.

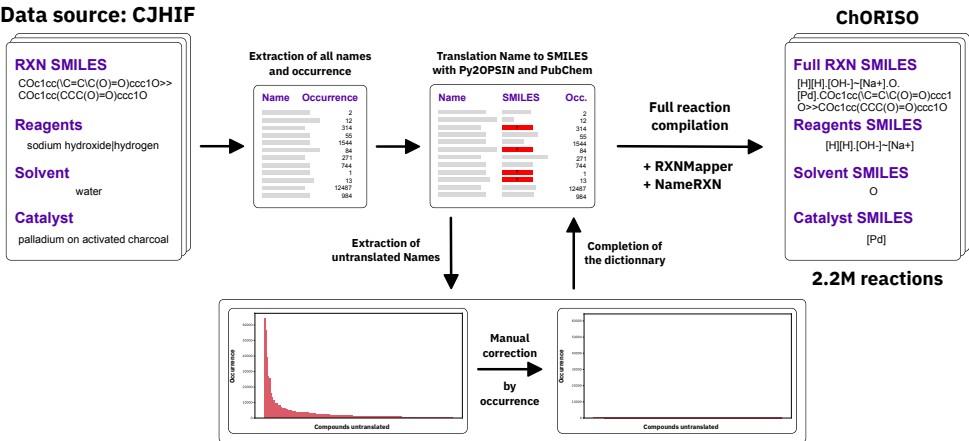

Figure 5: **Pipeline used for data curation.** The original compounds names in the CJHIF are extracted and listed by occurrence. Then, Py2OPSIN[40] and PubChem[72] are used to translate those names to SMILES strings. A manual correction of untranslated names is performed to get a more complete translation dictionary. Next, the original rows in the CJHIF are translated to SMILES to form full reaction SMILES. These reactions SMILES are then mapped using RXNMapper[74] and NameRxn[75].

The next step involves computing the atom mapping[76] for each reaction using both RXNMapper[74] and NameRxn[75].

The resulting dataset, CHORISO, contains 2'224'239 unique chemical reactions, encoded as reaction SMILES.

## A.2 Data analysis - Comparison with USPTO

The comparison between CHORISO and USPTO reveals differences in terms of reaction types and product distributions. The same cleaning pipeline was applied in both datasets in order to have a meaningful comparison. Reaction superclasses obtained with NameRxn for each dataset are shown in figure 2a. Dominant superclasses are different in both cases. Acylation and heteroatom alkylation and arylation are the most common classes in USPTO, representing more than 35% of the dataset. In the case of CHORISO, deprotection reactions are the most abundant (around 20%), followed by functional group interconversion (FGI) and heteroatom alkylation and arylation. The differences between classes in both datasets reflect the contrast in data sources. While USPTO samples were extracted from patents (with a higher proportion of pharma-related processes), CHORISO reactions come from academic journals, showing a high proportion of protective chemistry. Additionally, in both datasets more than 30% of the reactions cannot be assigned to a superclass (and therefore not shown in the previous figure plot).

As shown in Figure 2b and c, the distribution of product molecular weight in CHORISO shows longer tails than USPTO, with a better representation of lighter and heavier products which makes it more suitable for the OOD evaluations proposed here. Additionally, the proportion of products containing stereocenters in CHORISO is bigger than in USPTO. This diversity justifies its use as a new benchmarking dataset for reaction prediction models.

## A.3 Chemistry-specific metrics

Chemistry-specific metrics are designed as a refinement for top-n accuracy that considers a carefully selected subset of the test set, to test particularly the model's performance on challenging reactions with potential stereo- and regio-selectivity issues.

For the stereo-selective accuracy, reactions in the test set are filtered to include only those where the generation or inversion of a stereocenter occurs (Figure 6a). To do this, a reaction template with radius=0 is extracted from the queried reaction; if the reaction center in the product contains a '@' or '@@' character, it is used for model evaluation. Top-1 accuracy is then computed for this subset of reactions. A similar approach is followed for the regio-selective accuracy, however, the objective is now to identify reactions that could have undergone different regio-selectivity. For this, a reaction template with radius=1 is extracted from the queried reaction and then applied to the main reactants using RDKit RunReactants method. If several products are obtained, the reaction is used for model evaluation.

The resulting evaluation subsets contain 8440 (6.0%) and 13114 (9.4%) of the reactions in the test set of CHORISO's standard split, for stereo and regio, respectively. Despite the comparatively small size of these subsets with respect to the bigger testing set, these evaluations already permit an insightful decomposition of the model's performance across different chemically relevant aspects. Furthermore, stereo and regio accuracies represent a more realistic estimation of the model's ability to predict the results of a subset containing only stereoselective and regioselective reactions respectively.

## A.4 Out-of-distribution splits

After doing the splits, the resulting testing sets contain the following number of reaction SMILES:

- Product: 141130
- Random: 121144
- High PMW: 110056
- Low PWM: 90453

The product set contains reactions whose products are not contained in any of the reactions from the training set. The random split contains a random subset of the training reaction. The high PMW and low PWM splits contain reactions whose products are above 700 g/mol and below 150 g/mol

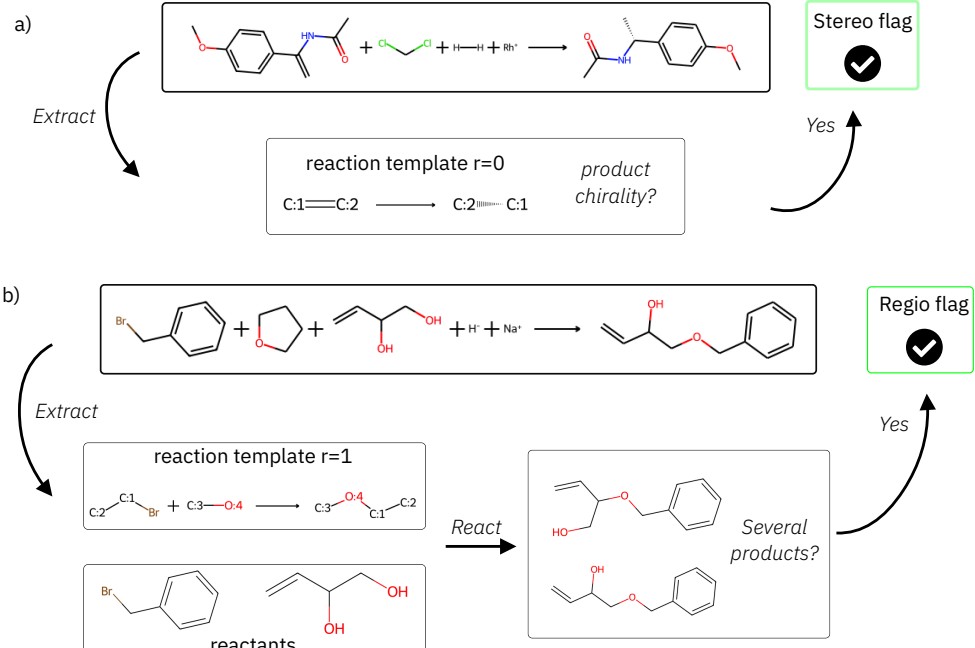

Figure 6: **Chemistry-specific metrics workflow**. a) Stereo-score checks if a chiral center is created or inverted in a reaction and flags it if True to compute the accuracy. b) Regio-score extracts the reaction template with radius=1 and applies it to the reactants. If more than one product is generated, it flags the reaction to compute the accuracy.

respectively, featuring an out-of-distribution shift on a non-relevant feature for reaction prediction as it is product molecular weight (PMW).

# B  Model benchmarking

In order to demonstrate the benefits of the proposed evaluation scheme with CHORISO, two SOTA models were selected for benchmarking: Molecular Transformer (MT) and Graph2SMILES (G2S). Detailed results are shown, including a comparison with models trained on previous benchmarks.

## B.1  Model performance

Table 2 displays the results of the evaluation over 4 metrics, across different data sources and split types (full benchmark).

| Data source | Split type | Model | Top-1 acc (%) | Top-2 acc (%) | Stereo-acc (%) | Regio-acc (%) |
|---|---|---|---|---|---|---|
| CHORISOv1 | Product | MT | 71.9 | 79.4 | 35.0 | 64.4 |
| | | G2S | 62.8 | 69.4 | 20.6 | 55.3 |
| | Random | MT | 72.6 | 80.2 | 37.7 | 64.9 |
| | | G2S | 62.9 | 69.6 | 22.2 | 54.5 |
| | low PMW | MT | 48.2 | 57.0 | 23.5 | 34.8 |
| | | G2S | 48.9 | 56.6 | 20.3 | 31.7 |
| | high PMW | MT | 26.7 | 29.8 | 12.8 | 24.7 |
| | | G2S | 33.6 | 37.4 | 17.0 | 32.8 |
| USPTO full* | Product | MT | 72.4 | 79.1 | 35.5 | 53.1 |
| | | G2S | 70.3 | 75.5 | 32.6 | 51.4 |
| | Random | MT | 73.5 | 80.1 | 40.3 | 54.9 |
| | | G2S | 70.7 | 76.0 | 34.4 | 51.7 |
| | low PMW | MT | 37.4 | 44.5 | 11.0 | 14.1 |
| | | G2S | 45.1 | 50.5 | 14.8 | 17.0 |
| | high PMW | MT | 28.1 | 30.2 | 6.4 | 13.3 |
| | | G2S | 37.5 | 40.3 | 9.6 | 17.8 |
| USPTO MIT** | Random | MT[42] | 88.6 | 93.7 | – | – |
| | | G2S[46] | 90.3 | – | – | – |

Table 2: **Model benchmarking results.** The performance of two models is shown, for each of two data sources, over the different types of data splits discussed here.
* Dataset curated using the same pipeline as for CHORISOv1. **Top-3 accuracies, top-2 accuracies were not reported in these studies.

As can be seen, the additional evaluation metrics and scenarios add new layers of insight to the comparison. In particular, MT is better at handling stereochemistry than G2S, while the latter largely outperforms in OOD scenarios. As highlighted in the main article, this information is instrumental not only for model selection for different use cases, but also serves to guide the research towards tackling specific objectives for different applications of the models. The results in Table 2 also shed light on the influence and importance of proper data splitting. In particular, this has a great influence on performance in the OOD settings, where performances are heavily affected by this choice of splitting type.

## B.2  Sustainability assessment

Sustainability assessments are central to our approach. More specifically, we focus on measuring $CO_2$ emissions and model traning and inference time . Table 3 shows the results for MT and G2S, across different data splits. For fair comparison, every model was trained for 200,000 training steps. Other hyperparameters for each model can be found on the benchmarking repository. All the sustainability metrics were tracked using the eco2AI package[69].

For purposes of visualization (Figure 1), both sustainability metrics were scaled between 0 and 100 to facilitate comparison between models and compatibility with the ranges of the other metrics. The resulting magnitudes therefore follow the same rational than top-k accuracy where a higher value normally means better performance. In particular, a formula of exponential decay was used for each metric. $CO_2$ production and Duration were scaled using eq. 1, with $k = 0.1$ for $CO_2$ and $k = 0.001$ for time.

$$S_x = 100e^{-kx} \tag{1}$$

The choice of $k$ was based on the scale of each metric in order to adjust it to the values on the main plot. These values are only used for visualization, and the ones that can be used for the analysis and comparison of models are the absolute values. Table 3 shows the sustainability metrics for the MT and G2S models trained on the CHORISO dataset. Sustainability metrics for the models trained on USPTO were not fully computed due to an error of the Eco2AI on the hardware. This shows .... and suggests that further improvement is needed in order to have a standardized energy consumption report.

| Model | Step | $CO_2$ (kg) | $CO_2$ scaled | Duration (h) | Duration scaled |
|---|---|---|---|---|---|
| MT | Training | 0.32 | 85.01 | 19.76 | 90.59 |
| | Inference | 0.04 | 97.86 | 3.14 | 98.44 |
| G2S | Training | 0.57 | 75.08 | 40.77 | 81.56 |
| | Inference | 0.05 | 97.76 | 3.71 | 98.16 |

Table 3: **Model training sustainability benchmarking.**

## C    Extra

All the results obtained for the CHORISO dataset were executed on a single GeForce RTX 3090 (24 GB) GPU. All results obtained for the USPTO dataset were executed on a single Tesla V100-PCIE (32GB) GPU. The model wrappers for training and evaluation can be found at `https://github.com/schwallergroup/CHORISO-models`.

