# OpenReview forum: "Holistic chemical evaluation reveals pitfalls in reaction prediction models"
_NeurIPS.cc/2023/Workshop/AI4Science — NeurIPS2023-AI4Science Oral_

### Official Review · Reviewer_cRNY · 2023-10-16
**A good evaluation paper**

**Rating:** 7
**Confidence:** 4

**Review:**

In this paper, the authors present a holistic analysis for the cutting-edge reaction prediction models. The author provides a new benchmark, ChORISOv1, along with multiple tailored splits to simulate chemically relevant scenarios, and provides several related metrics to better evaluate the model performance. The analysis reveals the limit for the current model as well as the evaluation protocols, and the provided benchmarks and metrics lead to re-thinking of the evaluation process of the reaction prediction model.

In summary, this is a good paper, I propose to accept this paper.

---

### Official Review · Reviewer_sQGn · 2023-10-22

**Rating:** 7
**Confidence:** 4

**Review:**

This paper introduces a new curated dataset ChORISOv1, along with a collection of metrics, splits that aims for a better evaluation of chemical evaluation.

Strength:
The proposed ChORISOv1 has several features making a good benchmarking dataset for reaction prediction. It constraints a high number of reactions, with a high ratio of C-C bond formation and function group interconverion. It also has a heavier-tailed distribution of molecular weight and number of stereocenters in products.
Two proposed metrics can better reflect the performance of the model: Regio- and stereo-accuracy are specifically designed to gauge the model’s ability to accurate predict the most reactive sites.

Cons:
Only two models are benchmarked. With such a limited number of models for comparison, it is hard to draw a meaningful conclusion.

The proposed new dataset and metrics are carefully designed and could be useful for advancing the reaction prediction field. Thus an accept is recommended.

---

### Meta-Review · Area_Chair_Babv · 2023-10-26

**Recommendation:** Accept (Oral)
**Confidence:** 4

**Metareview:**

The paper explores a very timely topic of out of distribution generalization in reaction prediction models. Indeed, currently there is no established benchmark to measure how well models generalize. The paper executes very well on this problem and introduces a large benchmark for reaction prediction based on previously extracted reactions from high-impact journals. Thank you for your submission and I am happy to recommend acceptance of the paper.

I have however one question to the Authors. How does the copyright work in this case? It seems that the original dataset is no longer available, see README.md of https://github.com/jshmjs45/data_for_chem. Is the submissions' paper licensed separately?